# Progression-Related Loss of Stromal Caveolin 1 Levels Mediates Radiation Resistance in Prostate Carcinoma via the Apoptosis Inhibitor TRIAP1

**DOI:** 10.3390/jcm8030348

**Published:** 2019-03-12

**Authors:** Julia Ketteler, Andrej Panic, Henning Reis, Alina Wittka, Patrick Maier, Carsten Herskind, Ernesto Yagüe, Verena Jendrossek, Diana Klein

**Affiliations:** 1Institute of Cell Biology (Cancer Research), University of Duisburg-Essen, University Hospital, Virchowstrasse 173, 45122 Essen, Germany; Julia.Ketteler@uk-essen.de (J.K.); Andrej.Panic@uk-essen.de (A.P.); Alina.Wittka@uk-essen.de (A.W.); Verena.Jendrossek@uk-essen.de (V.J.); 2Department of Urology and Urooncology, University of Duisburg-Essen, University Hospital, Essen, Hufelandstr. 55, 45122 Essen, Germany; 3Institute of Pathology, University of Duisburg-Essen, University Hospital, Hufelandstr. 55, 45122 Essen, Germany; henning.reis@uk-essen.de; 4Department of Radiation Oncology, University Medical Center Mannheim, Medical Faculty Mannheim, Heidelberg University, Theodor-Kutzer-Ufer 1-3, 68167 Mannheim, Germany; Patrick.Maier@medma.uni-heidelberg.de (P.M.); Carsten.Herskind@medma.uni-heidelberg.de (C.H.); 5Cancer Research Center, Division of Cancer, Imperial College London, Hammersmith Hospital Campus, London W12 0NN, UK; ernesto.yague@imperial.ac.uk

**Keywords:** Caveolin-1, TP53-regulated inhibitor of apoptosis 1, tumour stroma, tumour microenvironment, fibroblast, CAF, resistance, prostate cancer, radiotherapy

## Abstract

Tumour resistance to chemo- and radiotherapy, as well as molecularly targeted therapies, limits the effectiveness of current cancer treatments. We previously reported that the radiation response of human prostate tumours is critically regulated by CAV1 expression in stromal fibroblasts and that loss of stromal CAV1 expression in advanced tumour stages may contribute to tumour radiotherapy resistance. Here we investigated whether fibroblast secreted anti-apoptotic proteins could induce radiation resistance of prostate cancer cells in a CAV1-dependent manner and identified TRIAP1 (TP53 Regulated Inhibitor of Apoptosis 1) as a resistance-promoting CAV1-dependent factor. TRIAP1 expression and secretion was significantly higher in CAV1-deficient fibroblasts and secreted TRIAP1 was able to induce radiation resistance of PC3 and LNCaP prostate cancer cells *in vitro*, as well as of PC3 prostate xenografts derived from co-implantation of PC3 cells with TRIAP1-expressing fibroblasts *in vivo*. Immunohistochemical analyses of irradiated PC3 xenograft tumours, as well as of human prostate tissue specimen, confirmed that the characteristic alterations in stromal-epithelial CAV1 expression were accompanied by increased TRIAP1 levels after radiation in xenograft tumours and within advanced prostate cancer tissues, potentially mediating resistance to radiation treatment. In conclusion, we have determined the role of CAV1 alterations potentially induced by the CAV1-deficient, and more reactive, stroma in radio sensitivity of prostate carcinoma at a molecular level. We suggest that blocking TRIAP1 activity and thus avoiding drug resistance may offer a promising drug development strategy for inhibiting resistance-promoting CAV1-dependent signals.

## 1. Introduction

Cancer therapeutic resistance occurs through many different mechanisms, including specific genetic and epigenetic changes in the cancer cell itself and/or the respective microenvironment. The tumour stroma is now recognized as a key player in cancer cell invasiveness, progression and therapy resistance [1,2,3]. Activated fibroblasts (cancer associated fibroblasts, CAF) are capable of preventing cancer cell apoptosis and induce proliferation, as well as invasion, of surrounding cancer cells via direct stroma-tumour interactions by secreting extracellular matrix components, growth factors and matrix metalloproteinases, among others [4]. Although the exact mechanisms of fibroblast activation remain elusive, the activation or repression of specific genes or proteins within stromal cells has also been correlated with clinical outcome. Within that scenario, the membrane protein Caveolin-1 (CAV1) came into focus as it is highly expressed in many tumours and high CAV1 levels in tumour cells, as well as the downregulation of stromal CAV1, were shown to correlate with cancer progression, invasion and metastasis and thus, a worse clinical outcome [4,5]. Loss of stromal CAV1 can even be used as a prognostic marker, for example, in breast and prostate cancer patients [6,7,8,9]. Data on the CAV1-dependent epithelia-stroma crosstalk indicates that stromal CAV1 possesses tumour-suppressor properties, whereas loss of stromal CAV1 fosters malignant epithelial cell resistance by evading apoptosis [5,10]. Stromal loss of CAV1 is particularly prominent in epithelial prostate cancer, where loss of CAV1 in the stroma correlates with high Gleason score, presence of metastasis and pronounced resistance to chemotherapy and radiotherapy [6,8,11,12]. However, a detailed mechanism explaining how CAV1-deficient fibroblasts foster therapy resistance of malignant prostate cancer cells remains elusive. An improved understanding of the molecular basis of resistance will inevitably lead to the clinical assessment of rational drug combinations in selected patient populations.

An important mechanism by which cancer cells acquire drug resistance is by apoptosis evasion [3] and apoptosis inhibiting proteins have been described in both the development of cancer [13] and drug resistance [14]. TP53-regulated inhibitor of apoptosis 1 (TRIAP1, also known as p53-inducible cell-survival factor, p53CSV) is a small, 76 amino acids long, evolutionary conserved protein [15]. TRIAP1 was first characterized as a p53-inducible cell survival factor [16]. A genetic screen further identified TRIAP1 as a pathway-specific regulator of the cellular response to p53 activation [17]. Mechanistically, TRIAP1 modulates the apoptotic pathways through interaction with HSP70, inhibition of the interaction of cytochrome c with the apoptotic protease activating factor 1 and activation of the downstream caspase-9, thus resulting in increased resistance by inhibiting apoptosis and permitting DNA damage repair [15,16]. 

In this study, we aimed at determining the role of CAV1 alterations potentially induced by stromal CAV1-deficiency for the radio sensitivity of prostate cancer on molecular level and identified the apoptosis inhibitor TRIAP1 as a CAV1-dependent fibroblastic secreted factor, fostering radio resistance of malignant prostate epithelial cells.

## 2. Material and Methods

### 2.1. Reagents and Antibodies

Antibodies against CAV1 (N-20: sc-894) and XIAP1 (H-202: sc-11426) were from Santa Cruz (Santa Cruz, CA, USA), against CCND1 (92G2: #2978) and GFP (D5.1: #2956) from Cell Signalling Technology (Denvers, MA, Germany), against PCNA (PC10: GTX20029) from GeneTex (Irvine, CA, USA), against TRIAP1 from ProteinTech Group [15351-1-AP, (WB) Rosemont, IL, USA] and LSBio [LS-C346398-100, (Histology) Seattle, WA, USA), against SURVIVIN (NB500-201) from Novus Biologicals (Centennial, CO, USA) and against β-actin (clone AC-74, A2228) from Sigma-Aldrich (St. Louis, MO, USA). The rabbit anti human ASA antibody BE#3 was previously described [18] and the goat anti ASM antibody was kindly provided by Prof. K. Sandhoff (Bonn, Germany) [19].

### 2.2. Cell Culture Conditions

The human prostate cancer cell lines PC3, DU145 and LNCaP, the human skin fibroblast cell line HS5 and the human prostate fibroblast cell line WPMY-1 were from ATCC (Manassas, VA, USA) and cultured in RPMI Medium (Gibco, ThermoFisher, Waltham, MA, USA) supplemented with 10% foetal bovine serum and 100 U/mL Penicillin/Streptomycin under standard cell culture conditions (37 °C, 5% CO_2_, 95% humidity) and passaged every 3–4 days. CAV1 mRNA levels were down-regulated in indicated cells using shRNA technology as previously described [11,20,21]. For transient transfection of cells, human TRIAP1 cDNA with a C-terminal GFP-tag cloned into pCMV6-AC-GFP was used [15]. For selection of transfected cells, 500 µg/mL G418/Neomycin (Merck/Millipore, Darmstadt, Germany) was used.

### 2.3. Irradiation of Cell Cultures

Radiation was performed using the Isovolt-320-X-ray machine (Seifert-Pantak) at 320 kV, 10 mA with a 1.65 mm aluminium filter and a distance of about 500 mm to the object being irradiated [21]. The X-ray tube operated at 90 kV (~45 keV X-rays) and the dose rate was about 3 Gy/min [22].

### 2.4. Colony Formation Assay

The long-term survival assay was carried out by seeding 250 cells/well to 15.000 cells/well in a 6-well plate and irradiation at 0, 2, 4 and 6 Gy [11,21]. The plates were left to grow for 10 days into single colonies before they were fixed in 3.7% Formaldehyde (in PBS) and 70% Ethanol. Colonies were stained with 0.05% Coomassie Brilliant Blue for 1.5–3 h. Colonies (≥50 cells) were counted at fivefold magnification under the microscope.

### 2.5. Flow Cytometry Analysis

To measure and quantify the DNA-fragmentation (apoptotic sub-G1 population), as well as to quantify the cell cycle phases, cells were incubated for 30 min at RT with a staining solution containing 0.1% (*w*/*v*) sodium citrate, 50 μg/mL PI and 0.05% (*v*/*v*) Triton X-100 (*v*/*v*). Afterwards they were analysed by flow cytometry (FACS Calibur, Becton Dickinson, Heidelberg, Germany; FL-2).

### 2.6. Western Blotting

Generation of whole cell lysates was carried out by scraping cells off into ice-cold RIPA buffer (150 mmol/L NaCl, 1% NP40, 0.5% sodium-deoxycholate, 0.1% sodium-dodecylsulfate, 50 mmol/L Tris/HCl, pH 8, 10 mmol/L NaF, 1 mmol/L Na3VO4) supplemented with Protease-Inhibitor cocktail (Roche). After 2–3 freeze and thaw cycles the protein content of the lysates was measured by using DC™ Protein Assay (Bio-Rad). 50 µg to 100 µg of protein were loaded onto SDS-PAGE electrophoresis. Western blots were done as previously described [11,21] and the indicated antibodies were used to detect protein expression. 

### 2.7. Real-Time Reverse Transcription PCR (qRT-PCR)

RNA was isolated using RNeasy Mini Kit (74106, Qiagen, Hilden, Germany) according to the manufacturer’s instruction and as previously described [11,22]. Expression levels were normalized to the reference gene (β-actin; set as 1) and were shown as relative quantification. Specific primers were designed using Primer 3 [23] based on available NCBI nucleotide CDS sequences. Cross-reaction of primers was excluded by comparison of the sequence of interest with the NCBI database (Blast 2.2, U.S. National Centre for Biotechnology Information, Bethesda, MD) and all primers used were intron-spanning. PCR products are 200-300 bp in size. qRT-PCR was carried out using specific oligonucleotide primers (s sense, as antisense; TRIAP1s AGGATTTCGCAAGTCCAGAA, TRIAP1as GCTGATTCCACCCAAGTAT; TAGLNs TCCAGACTGTTGACCTCTTTGA, TAGLNas CCTCTCCGCTCTAACTGATGAT; ACTA2s GCCGAGATCTCACTGACTACCT, ACTA2as TGATGCTGTTGTAGGTGGTTTC; TGFB1s CCCACAACGAAATCTATGACAA, TGFB1as AACTCCGGTGACATCAAAAGAT; LAMP1s CCTGCCTTTAAAGCTGCCAA; LAMP1as CACCTTCCACCTTGAAAGCC; LAMP2s ACCACTGTGCCATCTCCTAC, LAMP2as TGCCTGTGGAGTGAGTTGTA; ACTINs GGCACCACACTTTCTACAATGA, ACTINas TCTCTTTAATGTCACGCACGAT) as previously described [11,22].

### 2.8. Conditioned Media

Cells were cultured in normal growth media until confluence. Cells were left non-irradiated or irradiated with 10 Gy, media were replaced and cells cultured in the presence of 0.5% foetal bovine serum for 48 h before collection of media. Control media were generated by incubating the same medium (containing 0.5% foetal bovine serum) without cells. Conditioned media were used as 1/1 mixture with normal growth medium [11,22]. 

### 2.9. Mouse Tumour Model

Mouse xenograft tumours were generated by subcutaneous injection of 0.5 × 10^6^ PC3 cells (+/−CAV1) either alone or mixed with 0.5 × 10^6^ WMPY-1 cells (+/−TRIAP1) onto the hind limb of male NMRI nude mice (total volume 50 µL) as previously described [11,21]. Animals of each experimental group received a single subcutaneous injection. For radiation therapy mice were anesthetized (2% isoflurane) and tumours were exposed to a single dose of 10 Gy ± 5% in 5 mm tissue depth (~1.53 Gy/min, 300 kV, filter: 0.5 mm Cu, 10 mA, focus distance: 60 cm) using a collimated beam with an XStrahl RS 320 cabinet irradiator (XStrahl Limited, Camberly, Surrey, Great Britain). Mouse experiments were carried out in strict accordance with the recommendations of the Guide for the Care and Use of Laboratory Animals of the German Government and they were approved by the Committee on the Ethics of Animal Experiments of the responsible authorities [Landesamt für Natur, Umwelt und Verbraucherschutz (LANUV), Regierungspräsidium Düsseldorf Az.8.87-50.10.37.09.187; Az.8.87-51.04.20.09.390; Az.84-02.04.2015.A586].

### 2.10. Human Tumour Tissue

Tissues from human prostate carcinomas were obtained during surgery according to local ethical and biohazard regulations. All experiments were performed in strict accordance with local guidelines and regulations. Resected tissue specimens were processed for pathological diagnostic routine in agreement with institutional standards and diagnoses were made based on current WHO and updated ISUP criteria [11,21]. All studies including human tissue samples were approved by the local ethics committee (Ethik-Kommission) of the University Hospital Essen (Nr. 10-4363 and 10-4051). Human tissue samples were analysed anonymously.

### 2.11. Immunohistochemistry and Immunofluorescence

Immunohistochemistry was performed on 4 µm slides of formalin-fixed and paraffin-embedded prostate tissues after performing a descending alcohol-series and incubation for 10 min to 20 min in target retrieval solution (Dako, Agilent, Santa Clara, CA, USA) [11]. After blocking of the slides with 2% NGS/PBS sections were incubated with primary antibodies o/n at 4 °C. Antigen were detected with horseradish-peroxidase conjugated secondary antibodies (1:250) and developed with DAB (Dako). Nuclei were counterstained using haematoxylin.

### 2.12. Statistical Analysis

If not otherwise indicated, data were obtained from 3 independent experiments with at least 2–3 mice each. Total mice numbers were stated in the figure legends. Statistical significance was evaluated by 1- or 2-way ANOVA followed by Tukey’s or Bonferroni multiple comparisons post-hoc test and set at the level of *p* ≤ 0.05. Data analysis was performed with Prism 5.0 software (GraphPad, La Jolla, CA, USA).

## 3. Results

### 3.1. Radioresistant (CAV1-Silenced) Fibroblasts Express and Secrete Anti-Apoptotic TRIAP1

We previously reported that CAV1-deficient fibroblasts foster radiation resistance of malignant prostate epithelial cells resulting in decreased apoptosis rates *in vitro* and *in vivo*, most likely via a paracrine mechanism of action [11]. Because we hypothesized that fibroblasts could allocate CAV1-dependent apoptosis inhibiting proteins to the tumour cells, we investigated the presence and expression levels of well-known resistance-associated anti-apoptotic proteins in stromal HS5 fibroblasts being either proficient [HS5(+)] for CAV1 or CAV1-deficient [HS5(-)] achieved by a shRNA-mediated knock-down (Figure 1). Of note, CAV1-silenced HS5 fibroblasts expressed significantly higher levels of TRIAP1 at both protein (Figure 1A) and mRNA level (Figure 1B). In addition, an increased CAV1-dependent TRIAP1 secretion was confirmed in cell culture supernatants of CAV1-silenced HS5 fibroblasts, which was accompanied by increased levels of lysosomal enzymes (acid sphingomyelinase, ASM and arylsulfatase A, ASA), which might be indicative for lysosomal exocytosis (Figure 1C).

Thus, increased expression and secretion of TRIAP1 by CAV1-silenced fibroblasts suggests that secreted TRIAP1 and then internalized by neighbouring prostate cancer cells, might account for the induced radiation resistance of these cells.

(A) Protein expression levels of apoptosis inhibiting proteins survivin, XIAP (X-linked inhibitor of apoptosis protein) and TRIAP1 were determined in CAV1-proficient [Cav1(+)] and CAV1-silenced [Cav1(-)] HS5 fibroblasts. Indicated proteins were analysed in whole protein lysates 96 h after radiation with 10 Gy by western blot analysis. Representative blots are shown. For TRIAP1 quantification, blots were analysed by densitometry and the respective signal was normalized to that from β-actin (n = 3–4 for each group). *p*-values were indicated: * *p* ≤ 0.05, ** *p* < 0.01 by one-way ANOVA followed by post-hoc Tukey’s test.

(B) qRT-PCR quantifications of TRIAP1 mRNA levels were performed 96 h post irradiation and shown as relative expression to β-actin mRNA. Data shown represent mean values ± SEM from 4 independent samples per group, each measured in duplicate. * *p* ≤ 0.05, ** *p* ≤ 0.01, by one-way ANOVA followed by post-hoc Tukey’s test.

(C) TRIAP1 and lysosomal enzymes (ASM, acid sphingomyelinase and ASA, arylsulfatase A) secretion were further determined in cell culture supernatants derived from CAV1-silenced HS5(-) or control transfected CAV1-expressing HS5(+) fibroblasts with or without radiation treatment (10 Gy) using western blot analysis. Equal protein amounts (100 μg) were loaded. Ponceau S staining of transferred proteins was included as loading control. 

### 3.2. Ectopic TRIAP1 Expression in Prostate Carcinoma Cells Induces Radiation Resistance

We previously have shown that cell culture supernatants of CAV1-silenced HS5 fibroblasts were able to induce radiation resistance of PC3 and LNCaP cells by decreased apoptosis [11]. We then investigated if the induced resistance of prostate cancer cells, after treatment with supernatants derived from CAV1-proficient or -deficient fibroblasts, led to higher TRIAP1 levels (not shown). However, no increased TRIAP1 levels were detectable in PC3, DU145 or LNCaP prostate carcinoma cells upon supernatants treatment most likely because the amount of tumour cell internalized TRIAP1 which was secreted from fibroblasts did not pass the threshold level of detection by western blot analysis. To provide the proof of principle that TRIAP1 mediates radiation resistance, the prostate cancer cells PC3 (p53 null), DU145 (p53 mutant) and LNCaP (p53 wild type) were transiently transfected with an expression vector encoding for human GFP-tagged TRIAP1 (Figure 2A). Empty vector transfected cells served as a control. Ectopic TRIAP1 expression resulted in decreased subG1 levels in PC3 and LNCaP cells 48 h after radiation with 10 Gy and thus increased resistance to radiation treatment. However, DU145 cells were not affected. Increased TRIAP1-levels were confirmed by western blot analysis (Figure 2B). Cell cycle analysis further revealed that ectopic TRIAP1 expression resulted in a slightly diminished G0/G1 subpopulation in PC3 cells upon radiation, while the proportion of cells in the G2/M phase increased (Figure 2C). The cell cycle of DU145 prostate carcinoma cells after TRIAP1 transfection was not affected upon radiation. Similar to PC3 cells, more TRIAP1-transfected LNCaP cells were in the G2/M phase after radiation as compared to control transfected cells. The proportions of respective cells in the S and <4n phase were rather low and not affected (not shown).

These results indicate that ectopic TRIAP1 expression mediates radiation resistance in a cell-type dependent manner and suggest that resistant prostate cancer cells will have an increased proliferation potential.

(A) Prostate cancer cells were transiently transfected with an expression vector encoding for human TRIAP1-GFP. Empty vector served as control. 24 h after transfection cells were irradiated with 0 or 10 Gy. The degree of apoptosis was quantified measuring the SubG1 fraction after radiation by flow cytometry analysis after additional 48 h of culture. Data shown represent mean values ± SEM from 4–5 independent samples per group measured in duplicates each. * *p* ≤ 0.05, by two-tailed students *t*-test.

(B) Efficiency of TRIAP1-GFP expressions as analysed by Western blots. Representative blots from 3-4 independent experiments are shown. β-actin is used as a loading control. As additional control (Ctrl) mock transfected cells, which underwent the transfection procedure without an expression vector were shown.

(C) Cell cycle analysis of TRIAP1-GFP transfected prostate cancer cell lines was performed using Nicoletti/PI staining and flow cytometry. Empty vector transfected cells served as control. Data represent mean values ± SEM from 3–5 independent samples per group measured in duplicates each. ** *p* ≤ 0.01, by two-way ANOVA followed by post-hoc Tukey’s test.

### 3.3. Generation of Stromal Prostate Fibroblasts with Stable TRIAP1 Expression

Prior to investigating whether TRIAP1 derived from a reactive tumour stroma might account for the radiation resistance observed in PC3 xenografts *in vivo* [11], we assessed the suitability of another fibroblast cell type, prostate fibroblasts (WPMY-1) derived from healthy donors, to more closely mimic the human situation in future *in vivo* experiments (Figure 3).

Compared to normal HS5 fibroblasts, WPMY-1 prostate fibroblasts expressed less endogenous CAV1-expression levels (Figure 3A). Quantitative Real Time RT-PCR analysis of TRIAP1 expression levels as well as of reactive fibroblasts markers (ACTA2 and TAGLN) and tumour-promoting EMT factor transforming growth factor β (TGFB1) in WPMY-1 fibroblasts (+/− XRT) confirmed the more reactive phenotype of WPMY-1 with a less pronounced CAV1-content and furthermore of irradiated WPMY-1 fibroblasts (Figure 3B). In line with previous findings [11], colony formation assays indicated that WPMY-1 fibroblasts with a reduced CAV1 content were more resistant to radiation (Figure 3C). To further investigate a potential TRIAP1-mediated radiation resistance of prostate carcinoma cells caused by the stromal compartment, we generated TRIAP1-overexpressing WPMY-1 fibroblasts via transfection of WPMY-1 with an expression vector encoding for human TRIAP1 tagged with GFP (Figure 3D). Stably transfected and TRIAP1-GFP-sorted cells (via flow cytometry) were successfully generated. Increased TRIAP1 expression was confirmed by western blot analysis. It is worth noting that TRIAP1-overexpression did not alter CAV1 expression levels (Figure 3D). Ectopic TRIAP1 expression resulted in a significant reduced subG1 population upon radiation, which confirmed the resistant phenotype of TRIAP1-GFP expressing prostate fibroblasts (Figure 3E). TRIAP1 secretion from TRIAP1-GFP-expressing cells was confirmed by western blot analysis from cell culture supernatants and revealed an increased secretion upon radiation (Figure 3F).

(A) CAV1 expression levels analysed by western blot in normal HS5 and prostate WPMY-1 fibroblasts, with or without radiation treatment with 10 Gy (96 h post irradiation). β-actin was included as loading control. Representative blots of at least three different experiments are shown.

(B) qRT-PCR quantifications of TRIAP1 mRNA levels, as well as reactive fibroblast markers, were performed 96 h post irradiation and shown as relative expression to β-actin mRNA. Data shown represent mean values ± SEM from 4-6 independent samples per group measured in duplicate each. * *p* ≤ 0.05, ** *p* ≤ 0.01, by two-way ANOVA followed by post-hoc Tukey’s test.

(C) Colony formation assay of HS5 and WPMY-1 cells. Following irradiation (0–8 Gy) cells were further incubated for 10 days. Data show the surviving fractions from three independent experiments measured in triplicates each (means ± SD). *** *p* ≤ 0.005, **** *p* ≤ 0.001 by two-tailed students *t*-test.

(D) The degree of apoptosis was quantified measuring the SubG1 fraction 48 h after radiation by flow cytometry. Data shown indicate mean values ± SEM from 3 independent samples per group measured in duplicates each. * *p* ≤ 0.05, by two-way ANOVA followed by post-hoc Tukey’s test.

(E) WPMY-1 prostate fibroblasts were transfected with a TRIAP1-GFP encoding plasmid or empty vector in the control, selected with G418 and sorted via flow cytometry to select GFP-expressing cells. Expression levels of TRIAP1-GFP and CAV1 were confirmed by western blot analyses, with or without 10 Gy irradiations. Band signal intensity was quantified by densitometry and normalized to that from β-actin. Data represent mean ± SEM from three independent experiments. *p*-values were indicated: ** *p* < 0.01; *** *p* ≤ 0.005 by two-way ANOVA followed by post-hoc Tukey’s test.

(F) TRIAP1-GFP secretion in cell culture supernatants derived from TRIAP1-GFP or control, transfected WPMY-1 fibroblasts with or without radiation treatment (10 Gy) determined by western blot analysis. CAV1, ASM and ASA secretion levels were also investigated. Equal protein amounts (100 μg) were loaded. Ponceau S staining of transferred proteins was included as loading control.

### 3.4. TRIAP1-Expressing Stromal Fibroblasts Mediate Radiation Resistance

Next, we asked whether fibroblastic tumour stroma-derived TRIAP1 accounts for an increased radiation resistance in PC3 xenograft tumours [11]. To mimic the human situation we performed subcutaneous transplantations onto the hind limb of NMRI nude mice by injecting CAV1-silenced PC3(-) tumour cells in combination with control-transfected or TRIAP1-GFP-expressing WPMY-1 prostate fibroblasts (Figure 4). Prostate xenografts were implanted onto the hind limb of NMRI nude mice and were irradiated locally with a single dose of 10 Gy when the tumour reached a size of about 100 mm3 (around day 3). Tumour growth was determined by measuring the tumour volume 3 times a week (Figure 4A). Either co-implantation with WPMY-1 cells, control or TRIAP1-transfected, did not change tumour growth. The tumour growth delay after radiation was significantly decreased in PC3(-)-derived tumours co-implanted with TRIAP1-expressing WPMY-1. These tumours showed a significantly increased growth after radiation treatment when compared to PC3(-)-derived tumours co-implanted with control-transfected WPMY-1 as demonstrated by the reduced time to reach a four-fold tumour volume (Figure 2B). Immunohistochemistry using the proliferation marker PCNA (proliferating cell nuclear antigen) antibody further confirmed an increase in the proliferation rate of PC3(-) xenografts when co-implanted with TRIAP1-GFP expressing fibroblasts and a significantly decreased sensitivity to radiation treatment (Figure 4C). Thus, in line with the *in vitro* results, TRIAP1 derived from stromal fibroblasts is able to induce radiation resistance of prostate tumours.

(A) PC3 CAV1(-) cells were subcutaneously injected with either TRIAP1-expressing WPMY-1 or control-transfected fibroblasts (0.5 × 10^6^ cells in total, ratio1/1) into the hind limb of NMRI nude mice. One set of animals from each group received a single tumour radiation dose of 10 Gy once its growth was easily detected (around day 3). Tumour volume was determined at indicated time points using a sliding calliper. Data are presented as mean +/− SEM from 3 independent experiments (26 mice in total: Ctrl 0 Gy n = 6; Ctrl 10 Gy n = 7; TRIAP1 0 Gy n = 6; TRIAP1 10 Gy n = 7).

(B) Tumour growth (*left panel*) and respective computed median growth delay (*right panel*) were determined as time (days) until a four-fold tumour volume was reached. *** *p* < 0.005; **** *p* < 0.001 by one-way ANOVA followed by post-hoc Tukey’s test.

(C) Immunohistochemical analysis of TRIAP1, PCNA and CAV1 in isolated PC3 xenograft tumours. Sections were counterstained using haematoxylin. Representative images are shown. Magnification 200×, scale bar 50 µm (left panel), magnification 400×, scale bar 20 µm (right panel).

### 3.5. Human Advanced Prostate Cancer Specimens Were Characterized by an Increased TRIAP1-Immunoreactivity Indicating Radiation Resistance

As loss of stromal CAV1 is paralleled by a radiation-resistance promoting reactive tumour stroma in human prostate tissue specimens [11,21], we decided to investigate TRIAP1 expression levels, as well as the respective stromal-epithelial TRIAP1 distribution, in human prostate tissue specimens by immunohistochemistry. TRIAP1 expression in prostate epithelial cells increased with higher Gleason scores, that is, lower tumour differentiation (Figure 5). Furthermore, stromal cells of tumour samples tended to be more intensively stained in cases with higher Gleason grade (Figure 5). These results indicate that increased, potentially fibroblast-derived, TRIAP1 has implications for prostate carcinoma progression and therapy resistance.

Paraffin-sections of human prostate cancers were stained for TRIAP1. Gleason grading scores were divided into low (Gleason Score ≥ 6, Grade group 1), intermediate (Gleason Score 7 (a/b), Grade groups 2 & 3) and high scores (Gleason Score ≥ 8, Grade groups 4 & 5). Asterisks mark stromal compartments and bold arrows point to epithelial structures. Sections were counterstained using haematoxylin. Representative images are shown. Magnification 200x. Right panel: higher magnification images: 400x. 

## 4. Discussion

Tumours are able to diversify their microenvironment and consequently the altered, more reactive tumour microenvironment can modulate the response of tumours to therapy treatment [24,25,26,27,28]. Herein, activated stromal cells and in particular activated fibroblasts/CAF can mediate therapy resistance of malignant epithelial cells in a CAV1-dependent fashion [4,5,29,30]. 

CAV1-dependent stromal-epithelial crosstalk in tumours with the potential to induce resistance includes processes such as autophagy or the ‘reverse Warburg effect’ [31,32]. For example, a more reactive stromal phenotype following a decrease of CAV1 expression by lysosomal degradation in fibroblasts was observed when cancer cells induced oxidative stress in the tumour-microenvironment [33]. In turn, downregulation of CAV1 in fibroblasts leads to increased oxidative metabolism in cancer cells, fostering cell resistance [29]. Importantly, extrinsic factors from the microenvironment and in particular from activated fibroblasts/CAF, may drive resistance in a non-tumour cell autonomous mechanism [34,35]. In line with these findings, we have recently shown that CAV1-deficient fibroblasts mediate radiation resistance of human prostate carcinoma cells *in vitro* and *in vivo* and that the decrease in cell death after radiation treatment is mediated though a paracrine mechanism of action [11]. However, the exact resistance-promoting effectors, as well as the role of CAV1-dependent fibroblast-derived factors, remained elusive. We therefore hypothesized that fibroblast-derived inhibitors of apoptosis proteins could mediate cell death resistance upon radiation. Here we show that TRIAP1 is highly expressed in stromal fibroblasts in a CAV1-dependent manner. *In vitro*, an ectopic expression of TRIAP1 leads to a cell specific increased radiation resistance in p53-deficient PC3 and p53-wildtype LNCaP prostate cancer cells, whereas p53-mutant DU145 cells do not gain any radiation resistance. Conformingly and mimicking the human situation more precisely, induced over-expression of TRIAP1 in human prostate fibroblasts leads to induced radiation resistance. Further on, TRIAP1-expressing stromal fibroblasts mediate radiation resistance *in vivo* when respective cells are co-implanted with CAV1-deficient PC3 tumour cells.

The underlying mechanism by which fibroblast-derived TRIAP1 is secreted and subsequently taken up by adjacent cancer cells and/or shuttled between the stromal and the tumour cells needs to be investigated further. TRIAP1 secretion in fibroblasts with a reduced CAV1-content is paralleled by the presence of lysosomal exocytosis related proteins and enzymes, such as ASM, ASA and LAMP proteins. This indicates that fibroblasts with a reduced CAV1 content bear a higher lysosomal exocytosis activity compared to fibroblasts containing normally high amounts of CAV1. It is known that the process and regulation of lysosomal exocytosis is largely changed upon tumour progression and in transformed cells [36]. Released lysosomal hydrolases, such as cathepsins D and B, play a role in tumour growth invasion and angiogenesis [37]. LAMP2 contributes to resistance, as the so called lysosomal cell death induced by anti-cancer drugs is decreased when LAMP2 is overexpressed in fibroblasts [38]. In addition, ASM is down-regulated in several carcinomas, for example, head and neck cancer and gastrointestinal carcinoma cancer cells, leading to a destabilized lysosomal environment in combination with an anti-apoptotic adaptation by decreased ceramide production [36]. Lysosomal exocytosis in cancer cells has been suggested to facilitate the entrapment and clearance of chemotherapeutics and provide an additional line of resistance [39].

As intrinsic drug resistance might be caused, at least in part, by factors secreted by the tumour microenvironment, it is thus imperative to dissect the tumour-microenvironment interactions which may reveal important mechanisms underlying drug resistance [35,39]. 

Interestingly, immunohistological analysis of TRIAP1 in advanced human prostate cancer reveals increased TRIAP1 immunoreactivity in the malignant epithelial cells of the more radioresistant higher Gleason grade adenocarcinomas. This highlights fibroblast-derived TRIAP1 as a potential candidate for future CAV1-mediated radiation response modulation. TRIAP1 is also involved in prostate cancer bone metastasis [40] and sensitivity to doxorubicin in breast cancer cells [15]. In ovarian cancer cells, increased TRIAP1 levels correlate with increased proliferation, a decrease in apoptosis and overall tumour progression [41]. TRIAP1 is also found to be upregulated in multiple myeloma [42], and, in patients with nasopharyngeal carcinoma, TRIAP1 overexpression correlates with a poor survival rate [43]. Experimental knockdown of TRIAP1, by expression of micro RNA miR-320b, is able to induce apoptosis by mitochondrial deregulating mechanisms, such as cytochrome C release and membrane potential alterations [15,43].

In summary, we have specified the role of CAV1 alterations potentially induced by CAV1-deficient and more reactive, stroma in radio sensitivity of prostate carcinoma at molecular level. We have identified apoptosis inhibitor TRIAP1 as a stromal-derived factor with the potential to induce cancer cell resistance. We suggest that blocking TRIAP1 activity and avoiding drug resistance may offer a promising drug development strategy to inhibit resistance-promoting CAV1-dependent signals.

## Figures and Tables

**Figure 1 jcm-08-00348-f001:**
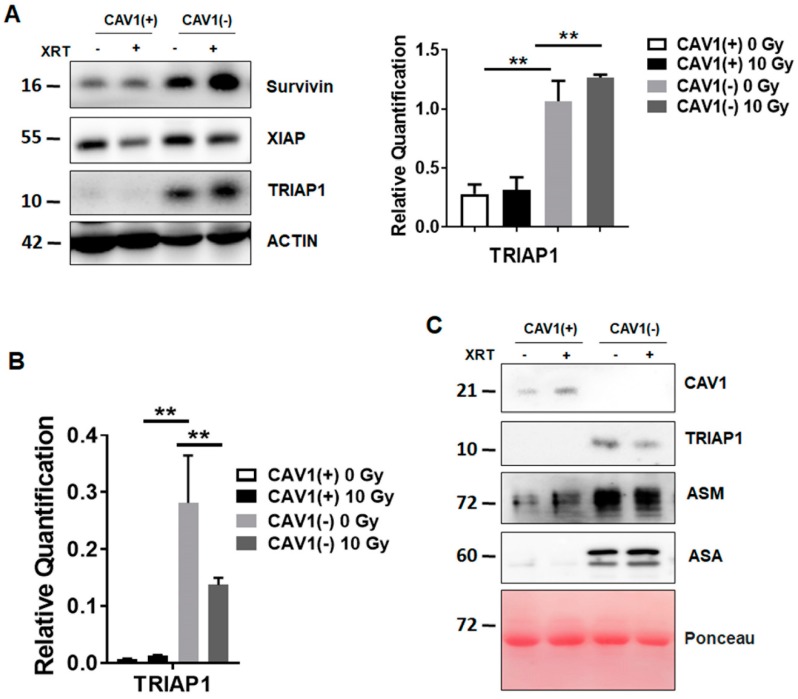
Radiation-resistant Caveolin-1 (CAV1)-silenced fibroblasts differentially express and secrete the apoptosis inhibiting protein TP53-regulated inhibitor of apoptosis 1 (TRIAP1).

**Figure 2 jcm-08-00348-f002:**
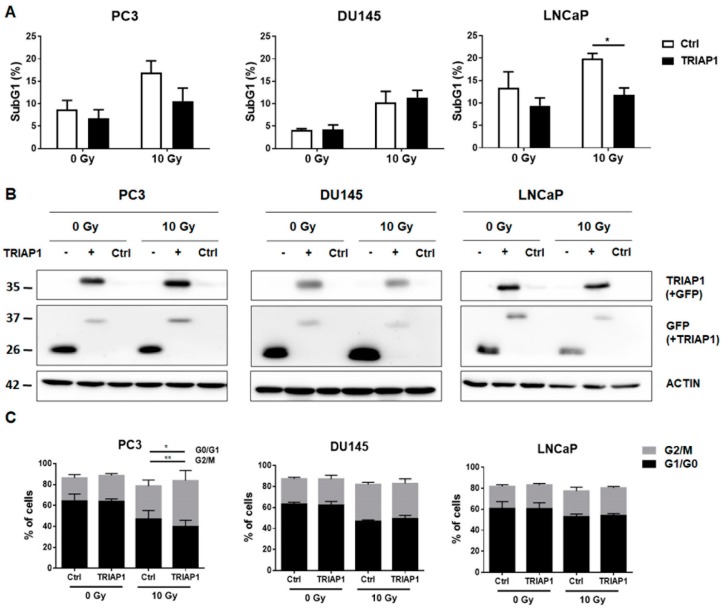
Ectopic TRIAP1 expression in prostate carcinoma cells results in radiation resistance.

**Figure 3 jcm-08-00348-f003:**
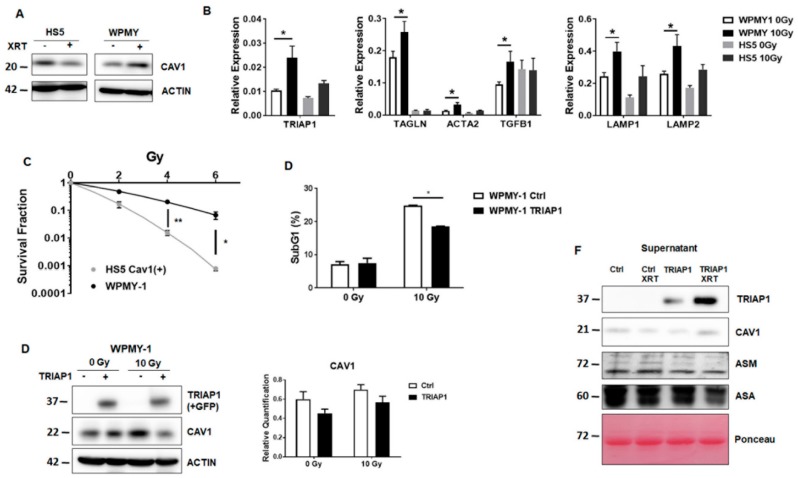
Characterization of the human prostate fibroblast cell line WPMY-1.

**Figure 4 jcm-08-00348-f004:**
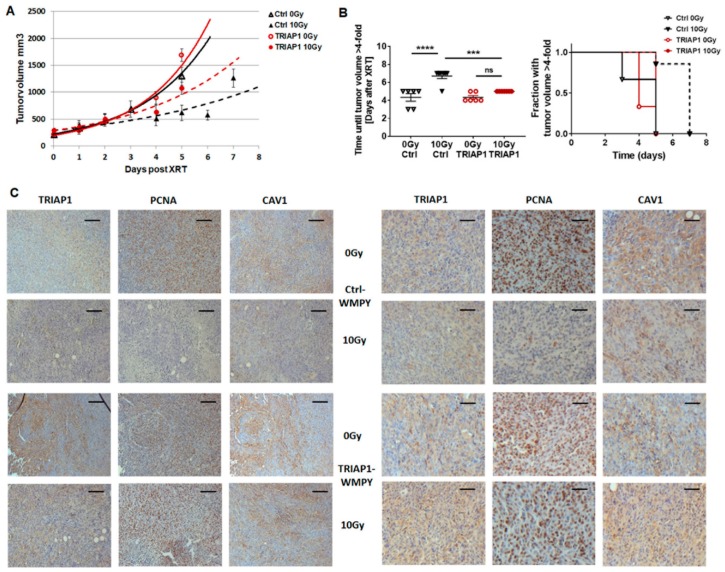
TRIAP1 expression in fibroblasts fosters radiation resistance in tumours derived from PC3 CAV1(-) cells.

**Figure 5 jcm-08-00348-f005:**
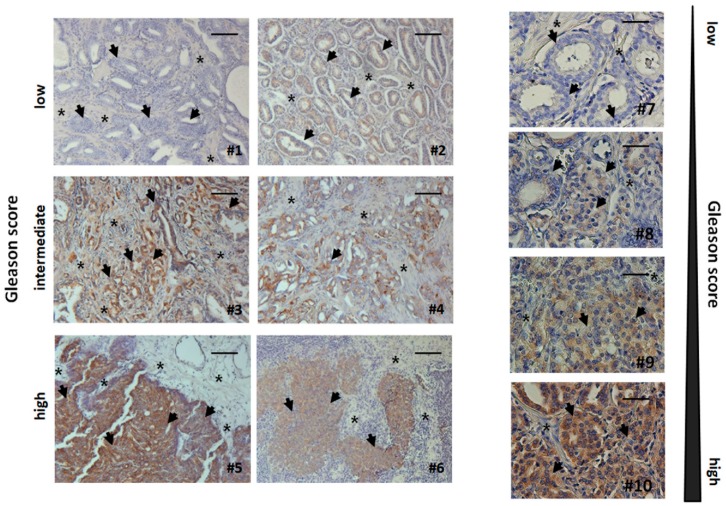
Immunohistochemical analysis of TRIAP1 expression levels in human prostate cancer tissues.

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
