# Peer review of "Progression-Related Loss of Stromal Caveolin 1 Levels Mediates Radiation Resistance in Prostate Carcinoma via the Apoptosis Inhibitor TRIAP1"

_jcm, 2019, doi:10.3390/jcm8030348_

Reviewer 1 Report

By using in vitro and in vivo approaches, Ketteler et al. describe a novel mechanism by which depletion of CAV1 from stromal fibroblasts drives radiation resistance through TRIAP1 upregulation. They also evaluated, by IHC, TRIAP1 expression in human advanced prostate cancer specimens providing a clinical significance. 

The manuscript is original, the experimental design as well as methods are appropriate, results are clearly described and appropriately discussed also in the context of the literature. Introduction provide sufficient background. 

Minor comments:

1) Some minor spell check or missing letters are present (e.g: Abstract, row 48 "mediatinG", Introduction, row 97 "aimed AT"

2) The manuscriptI would add just a reference related to an article describing the novel role of a protein whose function is the control of prostate cancer progression by eluding apoptosis in a mechanism dependent on p53-p21. (Antognelli C. et al. Mol Carcinog. 2017;56(9):2112-2126. doi: 10.1002/mc.22668).This can be inserted in the Introduction, lines 86-88: "An important ..... and apoptosis inhibiting proteins have been described in both the development of cancer  [HERE] and drug resistance [13], in order to strength how elusion of apoptosis is crucial also for prostate carcinogenesis. 

3) Sometimes Methods should be described in more details.

Author Response

Reviewer 1

By using in vitro and in vivo approaches, Ketteler et al. describe a novel mechanism by which depletion of CAV1 from stromal fibroblasts drives radiation resistance through TRIAP1 upregulation. They also evaluated, by IHC, TRIAP1 expression in human advanced prostate cancer specimens providing a clinical significance.

The manuscript is original, the experimental design as well as methods are appropriate; results are clearly described and appropriately discussed also in the context of the literature. Introduction provide sufficient background.

Answer

First, we would like to thank for the positive evaluation of our manuscript enabling us the re-submission of our revised manuscript. According to the reviewer’s suggestion, we addressed and corrected the critical points:

Minor comments:

Comment 1

1) Some minor spell check or missing letters are present (e.g: Abstract, row 48 "mediatinG", Introduction, row 97 "aimed AT"

Answer 1

This has been corrected. We carefully re-checked the manuscript for typing errors.

Comment 2

2) The manuscript would add just a reference related to an article describing the novel role of a protein whose function is the control of prostate cancer progression by eluding apoptosis in a mechanism dependent on p53-p21. (Antognelli C. et al. Mol Carcinog. 2017;56(9):2112-2126. doi: 10.1002/mc.22668).This can be inserted in the Introduction, lines 86-88: "An important ..... and apoptosis inhibiting proteins have been described in both the development of cancer  [HERE] and drug resistance [13], in order to strength how elusion of apoptosis is crucial also for prostate carcinogenesis.

Answer 2

According to the reviewers suggestion we included that important paper in the introduction.

Comment 3

3) Sometimes Methods should be described in more details.

Answer 3

We worked on the method section and described the methods in more Details.

Reviewer 2 Report

This manuscript reports potential factors underlying resistance to radiotherapy in prostate cancer. Specifically, TRIAP1, a secreted anti-apoptotic protein in the tumor microenvironment, might mediate therapeutic resistance to radiation. The authors showed that CAV1 loss in HS5 fibroblast cells correlated with increased expression of TRIAP1. Ectopic expression of TRIAP1 in prostate cancer cell lines PC3 and LNCaP but not DU145 appeared to confer radiation resistance. The authors went to show that the WMPY-1 prostate fibroblasts expressed lower amounts of CAV1 and appeared more “reactive” and radioresistant, although the expression levels of TRIAP1 were similar, compared to HS5 cells. The authors stably expressed TRIAP1-GFP fusion in WMPY-1 cells, which the authors claimed to be more radioresistant in vitro and to confer resistance of PC3 xenograft tumors to radiation therapy in vivo when PC3 cells and WMPY-1 cells expressing TRIAP1-GFP were co-transplanted into mice. These findings if validated with sufficient experimental rigor would be significant, as potential therapeutic strategies could be conceived to overcome radiation resistance.

Unfortunately, there are significant problems with the presented data as detailed below and must be satisfactorily addressed before this manuscript is acceptable for publication.

1.      The rationale for ectopic expression of TRIAP1-GFP in tumor cells is not convincing (Fig. 2). If TRIAP1 secreted from stromal cells works via a paracrine mechanism, the authors should use recombinant TRIAP1 at a range of concentrations to test whether it can indeed promote resistance to radiation. The level of resistance in transfected cells was insignificant in both PC3 and DU145 cells (Fig. 2A). What are the expression levels of endogenous TRIAP1 in these three prostate cancer cell lines?

2.      How TRIAP1 was detected by western blotting is confusing (Fig. 2B). What was the antibody used for detecting TRIAP1? Both endogenous TRIAP1 and TRIAP1-GFP fusion must be shown. What was the purpose of the control (Ctrl) in Fig. 2B? A molecular weight (MW) ladder for each western blotting panel must be provided.

3.      It is difficult to visualize the data shown in Fig. 2C. It is better to present separate graphs for each cell populations.

4.      CAV1 level appeared increased after radiation in WPMY-1 cells (Fig. 3A), which would be inconsistent with the authors’ contention that CAV1 downregulation promotes TRIAP1 expression.

5.      Which TGFB gene (TGFB1, 2, 3) was analyzed by qRT-PCR (Fig. 3B)?

6.      Fig. 3D: again, which antibody was used to detect TRIAP1? As in point #2 above, both endogenous TRIAP1 and TRIAP1-GFP fusion must be shown along with a MW ladder.

7.      The TRIAP1 quantification graph in Fig. 3D does not make sense. Which protein band was quantified? Was it TRIAP1-GFP? If so, what was quantified in the control?

8.      Fig. 3F: endogenous TRIAP1 along with other proteins as shown in Fig. 1C and a loading control that is not affected by radiation must be shown.

9.      Fig. 4A: what is the unfilled triangle? It is described that three independent experiments with a total of 26 mice in four experimental groups were used to generate the plot. How many mice were used in each experiment? Is this correct that two or 3 mice were used per group in each experiment?

10.  Fig. 4C: the image quality is poor. Images at higher resolution must be shown.

11.  Fig. 5: please label stromal and tumor cells on the IHC images. Is TRIAP1 diffusely localized in both stromal and tumor cells?

Minor points:

1.      The manuscript is written well in general, although there are numerous spelling and grammatical errors (e.g., “mediatin” should be “mediating” in the abstract; the sentence in lines 118 to 120 is incomplete …)

2.      Please provide vendor catalog # for all antibodies reported in the manuscript.

3.      Please provide the sequences of all qRT-PCR primers reported in the manuscript.

4.      Line 149: “probes” should be “lysates”

5.      Line 171: “0.5 x 106” should be “0.5 x 106

6.      Lines 45 and 388: “mouse tumors” should be “PC3 xenograft tumors”

7.      Were male mice used for the in vivo study?  

8.      Please provide MW ladder for all western blot panels in all figures.      

Author Response

Reviewer 2

This manuscript reports potential factors underlying resistance to radiotherapy in prostate cancer. Specifically, TRIAP1, a secreted anti-apoptotic protein in the tumor microenvironment, might mediate therapeutic resistance to radiation. The authors showed that CAV1 loss in HS5 fibroblast cells correlated with increased expression of TRIAP1. Ectopic expression of TRIAP1 in prostate cancer cell lines PC3 and LNCaP but not DU145 appeared to confer radiation resistance. The authors went to show that the WMPY-1 prostate fibroblasts expressed lower amounts of CAV1 and appeared more “reactive” and radioresistant, although the expression levels of TRIAP1 were similar, compared to HS5 cells. The authors stably expressed TRIAP1-GFP fusion in WMPY-1 cells, which the authors claimed to be more radioresistant in vitro and to confer resistance of PC3 xenograft tumors to radiation therapy in vivo when PC3 cells and WMPY-1 cells expressing TRIAP1-GFP were co-transplanted into mice. These findings if validated with sufficient experimental rigor would be significant, as potential therapeutic strategies could be conceived to overcome radiation resistance.

Answer

First of all we would like to thank for the positive evaluation of our manuscript enabling us the re-submission of our revised manuscript. According to the reviewer’s suggestion, we addressed and corrected the critical points (see below).

Unfortunately, there are significant problems with the presented data as detailed below and must be satisfactorily addressed before this manuscript is acceptable for publication.

Comment 1

1.     The rationale for ectopic expression of TRIAP1-GFP in tumor cells is not convincing (Fig. 2). If TRIAP1 secreted from stromal cells works via a paracrine mechanism, the authors should use recombinant TRIAP1 at a range of concentrations to test whether it can indeed promote resistance to radiation. The level of resistance in transfected cells was insignificant in both PC3 and DU145 cells (Fig. 2A). What are the expression levels of endogenous TRIAP1 in these three prostate cancer cells?

Answer 1

We absolutely agree with the reviewer that this is a very valuable point. We used the ectopic expression of TRIAP1-GFP in order to provide a proof of principle that TRIAP1 (and at this stage wherever it comes from) mediates radiation resistance. We were not able to detect endogenous TRIAP1 protein levels in these cell lines via Westernblot (WB) analysis. We already investigated a couple of commercial available TRIAP1 antibodies in the different analysis methods and found out that it is hard to detect TRIAP1. The listened antibodies worked well in WB and IHC. However, together with the putatively low endogenous TRIAP1 levels in the investigated prostate carcinoma cell lines, we have chosen TRIAP1-GFP to be able to detect the GFP tag in addition.

We already tried to use exogenous TRIAP1 “stimulation” of prostate carcinoma cells using cell culture supernatants derived from control and TRIAP1-GFP transfected WPMY-1 cells (see Figure below). Unfortunately, we were not able to detect any changes in the radiation response of the carcinoma cells, which is most likely due to the low TRIAP1(-GFP) levels in the cell culture supernatants. Together with the fact that a direct interaction between tumor and stromal cells might be necessary for the resistance promoting action of the fibroblasts, we argue that the proof of principle was already provided here. (i) Ectopic expression of TRIAP1-GFP in the different prostate carcinoma cells resulted in increased radio-resistance and (ii) prostate fibroblasts that over-expressed TRIAP1 mediated radiation resistance upon co-injection with PC3 cells.

Nevertheless, to address the concerns by the reviewer more precisely we already checked (and ordered one day ago) the availability of recombinant human TRIAP. We only found recombinant human HSPC132/TRIAP1 protein (#NBP2-22873 from Novus Biologicals) derived from E.coli, which might have the potential to function in stimulation /cell culture experiments. However, Novus (as well as the other companies) only reported the use of TRIAP1 as blocking peptide or SDS-PAGE. We will test whether recombinant TRIAP1 exogenously applied will mediate radiation resistance; however due to the period of the current revision we were not able to include these experiments right now.

Comment 2

2.     How TRIAP1 was detected by western blotting is confusing (Fig. 2B). What was the antibody used for detecting TRIAP1? Both endogenous TRIAP1 and TRIAP1-GFP fusion must be shown. What was the purpose of the control (Ctrl) in Fig. 2B? A molecular weight (MW) latter for each western blotting panel must be provided.

Answer 2

Again, we would like to thank the reviewer for this critical and important comment. To make the use of the right antibody clearer, we included now the MW markers. We refer also to the answer of comment 13. Ectopic expressed (as well as secreted) TRIAP1-GFP protein expression levels were detected either by TRIAP1 or by GFP antibodies. Vendor catalog # were now listened in the M&M section as recommended by the reviewer. As already stated before endogenous TRIAP1 in tumor cells was not detectable. TRIAP1 expression in HS5 fibroblasts either CAV1 proficient, or –deficient could be nicely detected by the listened TRIAP1 antibody. When WPMY-1 prostate fibroblasts were used, we switched to TRIAP1-GFP and then concentrated to detect TRIAP1 when fused to GFP with a TRIAP and GFP antibody (and the MW ~ 10kDa was removed/ were run out from the gels). The additional controls depict mock transfected cells, which underwent the transfection procedure without an expression vector. These controls were shown in Fig. 2B because they have been included on the gels all the time and we did not want to “cut out” the bands for the figure. We included that information now in the figure legend.

Comment 3

3.     It is difficult to visualize the data shown in Fig. 2C. It is better to present separate graphs for each cell populations.

Answer 3

Together with the fact that the SubG1 levels were already shown in Fig 2A, that the amount of cells in the S and >4n phase were not affected, and finally according to the reviewer’s suggestion, we now simplified Fig. 2C.

Comment 4

4.     CAV1 level appeared increased after radiation in WPMY-1 cells (Fig. 3A), which would be inconsistent with the authors’ contention that CAV1 downregulation promotes TRIAP1 expression.

Answer 4

We absolutely agree that sometimes membranes show the results not as clear as the quantifications; but of course, several biological and technical replicates were performed to generate, confirm and quantify the data presented. We apologize that the membrane presented was not perfect. We want to highlight that CAV1 expression levels in WPMY-1 cells were lower than the ones of HS5 cells which nicely correlates with the increased radiation resistance of the CAV1-low WPMY-1 cells in vitro (Fig. 3C). In addition, we want to stress the findings presented in Fig. 3D (now presented in Fig. 3E in the revised version), where we quantified the CAV1 protein levels in WPMY-1 cells with and without radiation. Although the WB presented here might indicate a minimal increase in CAV1 expression levels upon radiation, the quantification clearly shows that there is no increase.

Comment 5

5.     Which TGFB gene (TGFB1, 2, 3) was analyzed by qRT-PCR (Fig. 3B)?

Answer 5

The TGFB1 gene was analyzed by qRT-PCR (Fig. 3B). We included that missing information now in the figure.

Comment 6

6.     Fig. 3D: again, which antibody was used to detect TRIAP1? As in point #2 above, both endogenous TRIAP1 and TRIAP1-GFP fusion must be shown along with a MW latter.

Answer 6

We would like to refer to the answer of comment 2.

Comment 7

7.     The TRIAP1 quantification graph in Fig. 3D does not make sense. Which protein band was quantified? Was it TRIAP1-GFP? If so, what was quantified in the control?

Answer 7

We absolutely agree with the reviewer critics. There was obviously no protein quantified. We initially “quantified” the same area in the control lane as compared to the area derived from the TRIAP-GFP signal. In order to avoid any confusion and according to the reviewers critic, we removed now the diagram that showed the TRIAP1 quantification from the figure.

Comment 8

8.     Fig. 3F: endogenous TRIAP1 along with other proteins shown in Fig. 1C and a loading control that is not affected by radiation must be shown.

Answer 8

According to the reviewers suggestion we included now a loading control and (according to Fig. 1C) the other proteins in Fig. 3F.

Comment 9

9.     Fig. 4A: what is the unfilled triangle? It is described that three independent experiments with a total of 26 mice in four experimental groups were used to generate the plot. How many mice were used in each experiment? Is this correct that two or 3 mice were used per group in each experiment?

Answer 9

Unfortunately, there was a mistake in the symbols. The unfilled triangle belongs to the control group [PC3 Cav1(-) cells subcutaneously injected with control-transfected WMPY fibroblasts; Ctrl 0 Gy n= 6]. The symbols were now displayed uniformly. We further apologize for the unclear description. Data presented in tumor growth curves were obtained from 3 independent experiments with at least 2-3 mice each. “Total mice numbers were stated in the figure legends.” We included that information now in the M&M section: Statistical Analysis and in the respective figure legend (Fig. 4A): 26 mice were used in total: Ctrl 0 Gy n= 6; Ctrl 10 Gy n=7; TRIAP1 0 Gy n= 6; TRIAP 10 Gy n=7.

Comment 10

10.  Fig. 4C: the image quality is poor. Images at higher resolution must be shown.

Answer 10

According to the reviewers suggestion we included now higher resolution images.

Comment 11

11.  Fig. 5: please label stromal and tumor cells on the IHC images. Is TRIAP1 diffusely localized in both stromal and tumor cells?

Answer 11

We thank the reviewer for this important advice. According to the reviewer’s suggestion, we labeled stromal and tumor cells on the IHC images and included the information about the TRIAP1 localization in the result section.

Minor points:

Comment 12

1.     The manuscript is written well in general, although there are numerous spelling and grammatical errors (e.g., “mediatin” should be “mediating” in the abstract; the sentence in lines 118 to 120 is incomplete …)

2.     Please provide vendor catalog # for all antibodies reported in the manuscript.

3.     Line 149: “probes” should be “lysates”

4.     Line 171: “0.5 x 106” should be “0.5 x 106

5.     Lines 45 and 388: “mouse tumors” should be “PC3 xenograft tumors”

6.     Were male mice used for the in vivo study?

Answer 12

This has all been corrected according to the reviewer’s suggestion. We carefully re-checked the manuscript for typing errors. We included the requested information.

Comment 13

Please provide MW latter for all western blot panels in all figures.

Answer 13

According to the reviewer’s suggestion, we provide now MW latter for all western blot panels in all figures.

Round  2

Reviewer 2 Report

The authors addressed most of my concerns.

For Fig. 3F, is the MW (10) correct? Is GFP-TRIAP1 detected in the supernatant?

Please also provide the sequences of qRT-PCR primers used in the manuscript.  

Author Response

Comment 1 

For Fig. 3F, is the MW (10) correct? Is GFP-TRIAP1 detected in the supernatant? 

Answer 1 

Again, we would like to thank the reviewer for this important correction. We apologize for this mistake. Indeed, as described in the results section TRIAP1-GFP was detected in the supernatants. Accordingly, we corrected the MW in Fig. 3F.

Comment 2 

Please also provide the sequences of qRT-PCR primers used in the manuscript.  

Answer 2 

According to the reviewer’s suggestion, we provide now the qRT-PCR primers used in the manuscript (within the M&M section).